# VISTA Emerges as a Promising Target against Immune Evasion Mechanisms in Medulloblastoma

**DOI:** 10.3390/cancers16152629

**Published:** 2024-07-24

**Authors:** Natalia Muñoz Perez, Juliana M. Pensabene, Phillip M. Galbo, Negar Sadeghipour, Joanne Xiu, Kirsten Moziak, Rita M. Yazejian, Rachel L. Welch, W. Robert Bell, Soma Sengupta, Sonikpreet Aulakh, Charles G. Eberhart, David M. Loeb, Emad Eskandar, Deyou Zheng, Xingxing Zang, Allison M. Martin

**Affiliations:** 1Department of Pediatrics, Albert Einstein College of Medicine, 1300 Morris Park Avenue, Bronx, NY 10461, USA; juliana.pensabene@einsteinmed.edu (J.M.P.); phillip.galbo@roswellpark.org (P.M.G.J.); kirsten.moziak@einsteinmed.edu (K.M.); rita.yazejian@einsteinmed.edu (R.M.Y.); rachel.welch@einsteinmed.edu (R.L.W.); david.loeb@einsteinmed.edu (D.M.L.); eeskanda@montefiore.org (E.E.); deyou.zheng@einsteinmed.edu (D.Z.); xingxing.zang@einsteinmed.edu (X.Z.); 2Caris Life Sciences, Phoenix, AZ 85040, USA; nsadeghipour@carisls.com (N.S.); jxiu@carisls.com (J.X.); 3Department of Clinical Pathology & Laboratory Medicine, School of Medicine, Indiana University, 340 West 10th Street Fairbanks Hall, Indianapolis, IN 46202, USA; rb27@iu.edu; 4Department of Neurology & Neurosurgery, University of North Carolina at Chapel Hill, 170 Manning Drive, Chapel Hill, NC 27599, USA; ssengup@email.unc.edu; 5Department of Internal Medicine, West Virginia University, 64 Medical Center Drive, Morgantown, WV 26506, USA; sonikpreet.aulakh@hsc.wvu.edu; 6Department of Pathology, Johns Hopkins School of Medicine, 600 N Wolfe St., Baltimore, MD 21287, USA; ceberha@jhmi.edu

**Keywords:** medulloblastoma, immunotherapy, immune checkpoint inhibitor, tumor microenvironment, immune evasion, tumor-associated macrophages, T-regulatory cells, V-domain suppressor of T-cell activation

## Abstract

**Simple Summary:**

Immune checkpoint blockade has shown remarkable efficacy across various cancers but has failed to improve outcomes in patients with relapsed medulloblastoma (MB). While it is thought that the cold, immunosuppressive tumor microenvironment of MB accounts for this poor efficacy, the precise mechanisms contributing to immune suppression in this context remain unclear. In this study, we explore the immune landscape of MB using a previously unexplored syngeneic mouse model. Our study demonstrates that this model faithfully recapitulates the cold, immunosuppressive tumor microenvironment observed in human disease. Importantly, our research uncovers mechanisms employed by myeloid cells and tumor cells to evade immune detection while highlighting the therapeutic potential of targeting V-domain Ig Suppressor of T-cell Activation (VISTA), a novel inhibitory immune checkpoint, to enhance anti-tumor immunity.

**Abstract:**

Background: Relapsed medulloblastoma (MB) poses a significant therapeutic challenge due to its highly immunosuppressive tumor microenvironment. Immune checkpoint inhibitors (ICIs) have struggled to mitigate this challenge, largely due to low T-cell infiltration and minimal PD-L1 expression. Identifying the mechanisms driving low T-cell infiltration is crucial for developing more effective immunotherapies. Methods: We utilize a syngeneic mouse model to investigate the tumor immune microenvironment of MB and compare our findings to transcriptomic and proteomic data from human MB. Results: Flow cytometry reveals a notable presence of CD45^hi^/CD11b^hi^ macrophage-like and CD45^int^/CD11b^int^ microglia-like tumor-associated macrophages (TAMs), alongside regulatory T-cells (T_regs_), expressing high levels of the inhibitory checkpoint molecule VISTA. Compared to sham control mice, the CD45^hi^/CD11b^hi^ compartment significantly expands in tumor-bearing mice and exhibits a myeloid-specific signature composed of VISTA, CD80, PD-L1, CTLA-4, MHCII, CD40, and CD68. These findings are corroborated by proteomic and transcriptomic analyses of human MB samples. Immunohistochemistry highlights an abundance of VISTA-expressing myeloid cells clustering at the tumor–cerebellar border, while T-cells are scarce and express FOXP3. Additionally, tumor cells exhibit immunosuppressive properties, inhibiting CD4 T-cell proliferation in vitro. Identification of VISTA’s binding partner, VSIG8, on tumor cells, and its correlation with increased VISTA expression in human transcriptomic analyses suggests a potential therapeutic target. Conclusions: This study underscores the multifaceted mechanisms of immune evasion in MB and highlights the therapeutic potential of targeting the VISTA–VSIG axis to enhance anti-tumor responses.

## 1. Introduction

Medulloblastoma (MB) is the most common malignant embryonal brain tumor in children, with about 500 new cases in the United States per year [1,2]. MB tumors arise in the cerebellum and can grow rapidly, leading to cerebellar dysfunction and disruptions to cerebrospinal fluid flow [1,2]. This manifests in children as problems with gait, coordination, nausea, vomiting, and obstructive hydrocephalus [1,2]. The standard treatment paradigm for MB begins with surgical resection followed by radiation and/or chemotherapy, with specific regimens determined by a patient’s risk of progressive disease [1,2,3]. Historically, risk assessment relies on factors such as the patient’s age, the extent of tumor resection, and presence of metastasis [1,2,3]. While this stratification method yields positive outcomes for many patients, approximately one-third of high-risk patients still experience progressive or relapsed disease [1,2]. The limited stratification of patients into risk groups often overlooks the high intratumoral heterogeneity characteristic of MB [3,4]. Advancements in molecular profiling have led to the categorization of MB into distinct molecular subgroups: WNT-activated, SHH-activated, and non-WNT/non-SHH (consisting of Group 3/4) [3,5,6,7]. Each subgroup exhibits unique genomic landscapes and dysregulated molecular pathways, with the latter being notably more aggressive and associated with unfavorable treatment outcomes [3,5,6]. While this subgrouping has revolutionized the standard of care for patients with MB, relapsed disease remains a significant therapeutic challenge. Dominant-negative TP53 mutation and c-MYC amplification are frequently observed genomic alterations in relapsed disease and indicate aggressive disease behavior [4,8]. These tumors are generally refractory to traditional treatment modalities, regardless of the initial subgroup at diagnosis.

Early-phase clinical trials for patients with progressive or relapsed MB have evaluated the efficacy of FDA-approved immune checkpoint inhibitors, such as anti-PD-(L)1, which aim to enhance T-cell recognition and elimination of tumor cells (NCT02359565 and NCT03130959). While these immune checkpoint inhibitors have demonstrated efficacy across a variety of solid and hematological malignancies, they have not yielded success in treating MB patients [9,10]. The cold tumor microenvironment of MB, characterized by a lack of infiltrating T-cells and scarcity of PD-L1 expression, renders many of these immune checkpoint inhibitors ineffective [11,12,13]. However, the exact mechanisms behind poor T-cell infiltration have yet to be fully elucidated. Finding ways to enhance T-cell infiltration, activation, and persistence in the tumor can shift these tumor microenvironments from ‘cold’ to ‘hot’ and render them more responsive to immune checkpoint modulation [10].

Despite the scarcity of T-cells, tumor-associated macrophages (TAMs) have emerged as key cellular players in MB, comprising both tissue-resident microglia and bone marrow-derived macrophages (BMDMs) [14]. TAMs are remarkably plastic and adapt quickly to their surroundings. One possible mechanism that TAMs use to suppress T-cells is V-domain Ig Suppressor of T-cell Activation (VISTA), a novel B7 family immune checkpoint regulator predominantly expressed on hematopoietic cells, with enriched expression on myeloid cells, including microglia and macrophages [15,16,17]. This molecule has been implicated in suppressing T-cell activation, promoting T-cell and myeloid quiescence, and reprogramming myeloid cells towards an anti-inflammatory pro-tumoral state [15,18,19,20,21]. There has been an increased effort to target VISTA in solid tumors to activate the immune response against cancer cells [20,21,22,23,24,25]. Thus far, multiple preclinical models have shown that blocking VISTA deters tumor growth and improves T-cell effector function against tumors, including brain tumors such as gliomas [22,26,27,28,29]. Although VISTA-blocking antibodies are currently in Phase I clinical trials for solid tumor malignancies, this treatment remains untried in patients with brain tumors, and the effects of blocking VISTA in MB have not yet been explored.

Utilizing our previously established orthotopic syngeneic mouse model of MB harboring both a dominant-negative TP53 mutation and MYC overexpression, which lend it a clinically aggressive and anaplastic phenotype [30], we aimed to assess the extent to which the immune landscape resembles that of human MB. Additionally, we delved into the immunosuppressive mechanisms operating within this context. Our studies encompassed the characterization of myeloid polarization states and the immune checkpoints they express. Moreover, we determined the specific cellular and spatial patterns of VISTA expression within the tumor microenvironment. To our knowledge, this is the first study to set the stage for VISTA as an important mediator of immune suppression in MB and its potential as a valuable therapeutic target within this tumor microenvironment.

## 2. Materials and Methods

### 2.1. Cell Culture

All cells were maintained in a 37 °C humidified 5% CO_2_ incubator and manipulated in a BSL-2 certified tissue culture hood. A previously established mouse medulloblastoma cell line (mCB DNp53 MYC) which contains dominant-negative *TP53*, *c-MYC*, and an EGFP reporter, was used; this cell line originated in MD, USA, and was provided by Allison M. Martin [30]. Cells were grown in EF medium composed of DMEM/F12 (1:1) (+) L-Glutamine (+) 15 mM HEPES (Gibco, Waltham, MA, USA) supplemented with 2% B27 supplement, 5 μg/mL heparin, 20 ng/mL EGF, and 20 ng/mL FGF. Additionally, the human cell lines D283-MED and D425-MED were grown in MEM medium composed of MEM (+) L-Glutamine (Gibco) supplemented with 10% FBS, 1% sodium pyruvate, 1% MEM non-essential amino acids, and 1% penicillin/streptomycin. D283-MED was purchased from the American Type Culture Collection (ATCC, Manassas, VA, USA) and originated from NC, USA. D425-MED was a gift from Charles Eberhart and also originated in NC, USA. All cell lines were PCR-tested for mycoplasma every 6 months, and STR testing was performed annually on the human cell lines.

### 2.2. Animal Models

All animal experiments were conducted under the approval of the Albert Einstein College of Medicine Institute for Animal Care and Use Committee (IACUC, Bronx, NY, USA) protocol no. 00001069. C57BL/6J mice were obtained from Jackson Laboratories (Bar Harbor, ME, USA). Six-to-eight-week-old male and female mice were housed in a pathogen-free facility with access to standard chow and water. All surgical procedures were carried out in a BSL-2 certified laminar flow hood in the animal facility. To prepare the cells for implantation, mCB DNp53 MYC cells were collected, treated with StemPro Accutase Cell Dissociation reagent (LifeTech, Shenzhen, China), and resuspended in EF medium to yield a solution containing 100 cells per microliter (cells/µL). The mice were sedated with ketamine (100 mg/kg), and the posterior part of the head was shaved, sterilized, and locally anesthetized with 0.25% bupivacaine. A 1 cm midline craniocaudal incision was made in the scalp, exposing the lambdoid suture. An 18-gauge needle was used to drill a burr hole in the right cerebellum, positioned 1 mm posterior and 1 mm lateral to the lambda. Intracranial implantation of 300 mCB DNp53 MYC cells (3 µL) was performed by slowly injecting the cells with a 25-gauge Hamilton Syringe equipped with a sterile cut pipette tip to ensure the appropriate depth of 2 mm into the cerebellum, as previously described [30]. Sham control mice underwent an identical procedure but instead received 3 µL of EF medium. The incision was closed with surgical staples. Following the procedure, the mice were provided with MediGel Sucralose gels infused with carprofen (25 mg/kg +/− 3 mg/kg) in the days following the procedure. Staples were removed 10 days post-implantation. Mice were monitored for signs indicative of a moribund state, including ataxia, hunching, slowed movement, head tilt, and weight loss. The average time to disease onset in mice was 21.5 days.

### 2.3. Isolation of Tumor-Infiltrating Leukocytes

Once the mice exhibited a moribund state, tumors were harvested by gross dissection. Tumors were weighed and dissociated using the Neural Tissue Dissociation Kit (P) (Miltenyi, Tokyo, Japan) and transferred to a GentleMACS C-tube (Miltenyi) for further dissociation on the GentleMacs Octo Dissociator, with constant stirring at 37 °C for 30 min. Digestion was halted with the addition of 5 mL of cold MACS buffer (1X PBS, 0.5% BSA, 2 mM EDTA) and then strained through a 40 μm filter (Corning, Somerville, MA, USA). Myelin removal was performed by resuspending cells in a 30% Percoll gradient and spinning down at 700× *g* for 10 min at room temperature (RT). Red blood cell removal was performed using ACK lysis buffer (Quality Biological, Gaithersburg, MD, USA) for 5 min at RT. Either CD45 TIL microbead enrichment (Miltenyi) or CD11b microbead enrichment (Miltenyi) was performed. Both the positive bead-enriched fraction and negative flow-through were collected. Cells were counted and stained for multi-color spectral flow cytometry (see below). For the sham controls, the right cerebellar hemisphere was isolated 21 days following the surgical procedure, and the same downstream processing was performed as for the experimental samples.

### 2.4. Isolation of Spleens and Bone Marrow

Spleens and bone marrow from all tumor-bearing mice were collected. Bone marrow cells were isolated by flushing the femur with cold 1X PBS using a 25-gauge (BD Hypodermic, Franklin Lakes, NJ, USA) needle and centrifuging the cells at 300× *g* for 5 min. Spleens were collected and mashed through a 40-micron filter (Corning) and centrifuged at 300× *g* for 5 min. Red blood cell removal for spleens and bone marrow was performed using ACK lysis (Quality Biological, Gaithersburg, MD, USA) for 5 min at RT. Cells were washed with MACS buffer, counted, and stained for flow cytometry.

### 2.5. Multi-Color Spectral Flow Cytometry (CD45 Bead Enrichment)

CD45 TIL bead-enriched single-cell suspensions, spleen and bone marrow, were labeled with Zombie NIR Fixable Viability dye (Biolegend, San Diego, CA, USA, cat. no. 423105). Cells were then Fc-blocked with CD16/CD32 (Biolegend, cat. no. 101301). Cells were stained in a 1:1 solution of Brilliant Stain Buffer (BD Bioscience, Franklin Lakes, NJ, USA, cat. no. 563794) and 1X PBS containing the following fluorochrome-conjugated extracellular antibodies: CD45-AlexaFluor532 (ThermoFisher, Waltham, MA, USA, cat. no. 58-0451-80), CD11b-BUV395 (BD Bioscience, cat. no. 565976), CD11c-Brilliant Violet 570 (Biolegend, cat. no. 117331), CX3CR1-AlexaFluor700 (Biolegend, cat. no. 149035), Tmem119-PerCPeFluor710 (ThermoFisher, cat. no. 46-6119-80), CD206-Brilliant Violet 421 (Biolegend, cat. no. 141717), CD3-Brilliant Violet 650 (Biolegend, cat. no. 100229), CD4-Brilliant Violet 605 (Biolegend, cat. no. 100547), CD8-BUV805 (ThermoFisher, cat. no. 368-0081-80), CD19-Brilliant Violet 510 (Biolegend, cat. no. 115545), NK1.1-Pacific Blue (Biolegend, cat. no. 108721), VISTA-APC (Biolegend, cat. no. 150205), and B7H3-PECy7 (Biolegend, cat. no. 135613). Intracellular fixation was performed using the FOXP3/Transcription Factor staining kit (ThermoFisher). Cells were permeabilized and stained with IFNy-PECF594 (BD Bioscience, cat. no. 562333) and FOXP3-PE (ThermoFisher, cat. no. 14-5773-80) diluted in permeabilization buffer. Cells were resuspended in MACS buffer, and data was collected on the Aurora Cytek maintained by the AECOM Flow Core Facility. Immune populations were gated in FlowJo v.10.10.0 (Appendix A). All gating was based on fluorescence minus one (FMO) controls.

### 2.6. Multi-Color Spectral Flow Cytometry (CD11b Bead Enrichment)

To more closely look at myeloid cells from tumor-bearing or sham control mice, CD11b bead-enriched single-cell suspensions were labeled with Zombie NIR Fixable Viability dye (Biolegend, cat. no. 423105). Cells were then Fc-blocked with CD16/CD32 (BD Biosciences). Cells were stained in a 1:1 solution of Brilliant Stain Buffer (BD Bioscience, cat. no. 563794) and 1X PBS containing the following extracellular fluorochrome-conjugated antibodies: CD45-AlexaFluor532 (ThermoFisher, cat. no. 58-0451-80), CD11b-BUV395 (BD Bioscience, cat. no. 565976), CD11c-Brilliant Violet 570 (Biolegend, cat. no. 117331), CD3-Pacific Blue (Biolegend, cat. no. 100213), F4/80-PerCPVio700 (Miltenyi cat. no. 130-118-466), Arg1-AlexaFluor700 (ThermoFisher, cat. no. 56-3697-80), PDL1-Brilliant Violet 605 (Biolegend, cat. no. 124321), VISTA-APC (Biolegend, cat. no. 143709), CD40-PECy7 (Biolegend, cat. no. 124621), CD163- PE/Dazzle594 (Biolegend, cat. no. 155315), CD68-Brilliant Violet 711 (Biolegend, cat. no. 137029), Tim3-PE (Biolegend, cat. no. 134003), CD80-Brilliant Violet 650 (Biolegend, cat. no. 104713), and CTLA4-Brilliant Violet 421 (Biolegend, cat. no. 106311).

### 2.7. Flow Cytometry

For evaluation of VSIG3 and VSIG8 expression on human and mouse cell lines, both D283-MED and mCB DNp53 MYC were collected and stained with Live/Dead Fixable Aqua (ThermoFisher, cat. no. L34965). Cells were washed and then stained with either rabbit anti-VSIG3 (R&D, Santa Clara, CA, USA, cat. no. MAB11226-SP), rabbit anti-VSIG8 (ThermoFisher, cat. no. PA5-51157), or rabbit IgG isotype control (R&D, cat. no. AB-105-C) for 15 min at RT in the dark. Cells were washed and then incubated with goat anti-rabbit-APC IgG (H+L) cross-absorbed secondary antibody (ThermoFisher, cat. no. A-10931) for 15 min at RT in the dark. Cells were fixed with 2% paraformaldehyde and analyzed on the BD LSRII maintained by the AECOM Flow Core Facility. Gating was performed in FlowJo v10.10.0 and based on fluorescence minus one (FMO) controls.

### 2.8. Western Blot Analysis

Protein was isolated from D283-MED, D425-MED, and mCB DNp53 MYC cells using an extraction buffer (RIPA buffer, 0.01 M NaF, 100 mM sodium vanadate, 1X Sigma protease inhibitor cocktail, 0.1 M PMSF), followed by protein quantification with the DC Protein Assay Kit (BioRad, Hercules, CA, USA) and measurement of absorbance on the SpectraMax M5e at 750 nm. Each sample received 20 µg of respective protein and was run on TGX precast gels on the Mini Protean TetraCell system. Protein was transferred to a PVDF membrane using the Biorad TransBlot Turbo transfer system. The membrane was blocked in 5% milk and incubated with primary antibodies overnight at 4 °C: Rabbit anti-VSIG3 (R&D, cat. no. MAB11226-SP), Rabbit anti-VSIG8 (ThermoFisher, cat. no. PA5-51157), or Sheep anti-VISTA (R&D, cat. no. AF-7005-SP). Next day, the membrane was washed with TBST and incubated with Goat anti-Rabbit IgG (H+L), Peroxidase (Vector Laboratories, Burlingame, CA, USA, cat. no. PI-1000-1), or Donkey anti-Sheep IgG HRP-conjugated (R&D, cat. no. HAF016) for 1 h at RT, followed by three consecutive TBST washes. Chemiluminescence imaging was performed on the Biorad ChemiDoc Touch imaging system.

### 2.9. Immunohistochemistry

Tumors were harvested and fixed in FormaFixx (ThermoFisher) at 4 °C before submission to the Histology Core for paraffin embedding and sectioning. Sectioning was performed every 5 microns. Briefly, serial sections were deparaffinized and incubated for 10 min in 3% hydrogen peroxide. Antigen retrieval was performed in 1X DAKO (Agilent, Santa Clara, CA, USA) for 30 min using an electric steamer (Oster, Boca Raton, NW, USA), and then the samples were allowed to cool at RT for 30 min. Fc blocking was performed for 30 min at RT, followed by blocking with 2.5% Normal Horse Serum (Vector Laboratories) for 30 min at RT. Slides were incubated overnight at 4 °C with primary antibody: rabbit anti-mouse CD11b (Cell Signaling Technology, Danvers, MA, USA, cat. no. 93169S), rabbit anti-mouse VISTA (Cell Signaling Technology, cat. no. 54979T), anti-mouse CD3 (Cell Signaling Technology, cat. no. 78588T), rabbit anti-mouse FoxP3 (Cell Signaling Technology, cat. no. 12653), or rabbit mAb IgG isotype control (Cell Signaling Technology, cat. no. 3900S). The following day, slides were washed with TBST and incubated with ImPRESS HRP horse anti-rabbit secondary antibodies (Vector Laboratories). Slides were developed with DAB peroxidase substrate (Vector Laboratories) for up to 90 seconds. Slides were counterstained with hematoxylin (Vector Laboratories), dehydrated, and mounted with Surgipath Micromount (Leica, Wetzlar, Germany). Slides were allowed to dry, and then imaging was performed on the EVOSS microscope. White balancing was performed using Adobe Photoshop. Scale bars were incorporated in Fiji by aligning pixels to microns through calibration using a cell-counter slide.

### 2.10. Tumor Cell and T-Cell Co-Cultures

Naïve splenic CD4 T-cells were isolated from a C57BL/6J wildtype mouse using the EasySep^TM^ Mouse CD4+ Isolation Kit (StemCell Technologies, Vancouver, Canada, cat. no. 19852). Following isolation, 80,000 CD4 T-cells were plated in a 96-well V-plate and stained with CellTrace-Violet (ThermoFisher) for 30 min at RT. T-cells were resuspended in 100 µL of RPMI medium supplemented with 10% FBS, 1% Pen/Strep, 1% HEPES, 1% sodium pyruvate, 1% NEAA, 55 uM B-mercaptoethanol, and 0.45% D-glucose, in addition to 1 ng of recombinant mouse IL2 (BioLegend, San Diego, CA, USA, cat. no. 575402) and 1:1 CD3/CD28 Dynabeads (Gibco, cat. no. 11-452-D). T-cells were then co-cultured with 100 µL of either RPMI medium, EF medium, conditioned medium (isolated by filtering cultured supernatant through a Nalgene 0.2 µm syringe filter), or increasing numbers of mCB MYC DNp53 tumor cells (T-cell/mCB DNp53 MYC ratios: 80:1, 40:1, and 20:1). Negative controls included CD4 T-cells with no CD3/CD28 bead activation. Co-cultures were transferred to a 96-well U-plate. After 5 days of incubation, cells were stained with L/D-Zombie NIR (Biolegend, cat. no. 423105), CD4-Brilliant Violet 605 (Biolegend, cat. no. 100547), and VISTA-APC (Biolegend, cat. no. 143709) and analyzed on the Cytek Aurora. Gating was performed using FlowJo, where CellTrace Violet dilution was measured as a function of T-cell proliferation. To see if blocking VISTA could rescue T-cell proliferation, CD4 T-cells were co-cultured with mCB MYC DNp53 tumor cells at a ratio of 20:1 for 5 days, as described above. Addition of either 0, 0.2, 2, or 20 µg/mL of InVivoMab anti-mouse VISTA blocking Ab (BioXCell, Lebanon, NH, USA, cat. no. BE0310) or 20 µg/mL of an InVivoMab polyclonal Armenian Hamster IgG Isotype control (BioXCell, cat. no. BE0091) was performed in technical triplicates over 6 biological replicates. After 5 days, cells were stained with L/D-Zombie NIR (Biolegend, cat. no. 423105), CD4-Brilliant Violet 605 (Biolegend, cat. no. 100547), and VISTA-APC (Biolegend, cat. no. 143709) and analyzed on the Cytek Aurora maintained by the AECOM Flow Core Facility. Gating was performed using FlowJo, where CellTrace Violet dilution was measured as a function of T-cell proliferation.

### 2.11. Caris Molecular Profiling Analysis

Biomarker data acquired from the Caris database contained records of 81 patients whose medulloblastoma tumors underwent RNA (whole-transcriptome) and DNA (next-generation) sequencing at Caris Life Sciences (Phoenix, AZ, USA). Of these patients, 53 were aged 30 or under, forming the primary focus of the analysis. One patient was excluded from the analysis due to frontal lobe primary tumor location. This study was conducted in accordance with the guidelines of the Declaration of Helsinki, the Belmont Report, and US Common Rule. In compliance with policy 45 CFR 46.101(b), this study was performed utilizing retrospective, deidentified clinical data, thereby qualifying for IRB exemption, obviating the need for patient consent. To evaluate mRNA expression (WTS), the Qiagen RNA FFPE tissue extraction kit was used on tumor specimens (with a minimum of 10% tumor content for enrichment and extraction of RNA), and the RNA quality and quantity were determined via Agilent TapeStation. Biotinylated RNA baits were hybridized to the synthesized and purified cDNA targets and the bait–target complexes were amplified in a post-capture PCR reaction. The Illumina NovaSeq 6500 was used to sequence the whole transcriptomes from patients to an average of 60 M reads (Illumina, San Diego, CA, USA). Raw data were demultiplexed by the Illumina Dragen BioIT accelerator, trimmed, and counted, and PCR duplicates were removed and aligned to the human reference genome hg19 by STAR aligner. For transcription counting, transcripts per million (TPMs) were generated using the Salmon expression pipeline.

### 2.12. Nanostring GeoMx^®^ Digital Spatial Profiling

A single formalin-fixed paraffin-embedded slide containing multiple embryonal human tumor cores in the form of a tissue microarray was stained with 4 immunofluorescent markers, including DAPI, synaptophysin, CD45, and Iba1, and 77 additional proteins from NanoString GeoMx^®^ Digital Spatial Profiling curated panels, including Immune Cell Profiling, IO Drug Target, Immune Activation Status, PIK/AKT, and Immune Cell Typing panels (NanoString, Seattle, WA, USA). Twelve samples representing distinct tumors were selected based on tissue integrity: four SHH-activated medulloblastoma samples, five non-WNT/non-SHH medulloblastoma samples, and three medulloblastoma samples of indeterminate subgroups. Analysis and selection were performed in real time based on the appearance of a paired H&E slide. Rare cell masking software identified Iba-1-positive geometric regions within the tumor cores and was applied to selected samples. Photocleavable linkers were activated to collect protein specifically from IBA-1-positive and -negative regions based on masking.

### 2.13. Statistical Analysis

Results are presented as mean values +/− standard error of the mean (SEM). Statistical tests for in vitro and in vivo experiments were conducted in GraphPad Prism v10. In all cases, a *p*-value of less than 0.05 was considered statistically significant. For heatmap generation of in vivo murine data and in situ human data, markers were considered significantly different between compared groups with a log2 fold change less than −0.5 or greater than 0.5 and a *p*-value less than 0.05. For Caris Life Sciences transcriptomic data, continuous variables were compared using non-parametric tests, including Mann–Whitney U tests. Adjustments for multiple comparisons were applied using the Benjamini–Hochberg method to avoid type I errors. An adjusted *p*-value (q-value) of <0.05 was considered a significant difference. Specifically, for NanoString GeoMx^®^ Digital Spatial Profiling data analysis, protein expression was transformed with the natural log. Differentially expressed proteins were analyzed with the limma package (v3.48.3) [31].

## 3. Results

### 3.1. A Syngeneic Murine Medulloblastoma Model, mCB DNp53 MYC, Is Representative of the Cold Tumor Microenvironment

To investigate the immune landscape of MB in an immune-competent model, we orthotopically implanted mCB DNp53 MYC cells into the right cerebellum of C57/BL6J mice. Once moribund, tumors were harvested and infiltrating immune cells were CD45 bead-enriched and immune-phenotyped by multi-color flow cytometry (Figure 1a). The presence of GFP+ tumor cells was confirmed in the CD45-negative flow-through (Appendix A). Within the CD45-positive bead-enriched compartment, we observed a significantly higher percentage of CD11b+ myeloid cells within the tumor microenvironment (percent mean ± SEM = 55 ± 5.66%) as compared to CD4+ T-cells (percent mean ± SEM = 9.41 ± 1.97%) and few to no CD8+ T-cells (percent mean ± SEM = 0.204 ± 0.155%) (*n* = 6 tumor-bearing females) (Figure 1b). Furthermore, there was a notable scarcity of B-cells and NK-cells (Appendix A). Spleens were used as technical controls to ensure appropriate lymphocyte staining (Appendix A). Within the CD11b+ myeloid compartment, cells could be subdivided into two sub-populations: CD45^intermediate(int)/^CD11b^intermediate(int)^ (percent mean ± SEM = 29.50 ± 4.51%) and CD45^high(hi)/^CD11b^high(hi)^ (percent mean ± SEM = 25.50 ± 2.79%), which have previously distinguished microglia from bone marrow-derived macrophages, respectively (Figure 1c) [14]. Notably, both CD45^int^/CD11b^int^ and CD45^hi^/CD11b^hi^, which were non-dendritic (CD11c-), strongly co-expressed both VISTA and MHCII (Figure 1c). In stark contrast, CD45+CD11b+CD11c- cells from the bone marrow of tumor-bearing mice exhibited low co-expression of VISTA and MHCII, indicating that upregulation of these two markers was tumor-specific (Appendix A). Additionally, neither subpopulation expressed the microglia marker Tmem119, as has been observed in other MB models [32]. Within the CD4 T-cell compartment, we observed that a large percentage of CD4+ T-cells were FOXP3+ T_regs_ (percent mean ± SEM = 22.95 ± 4.968%) (Figure 1b). Upon further analysis of these immune subpopulations, we found that CD45^hi^/CD11b^hi^ macrophage-like TAMs, CD45^int^/CD11b^int^ microglia-like TAMs, and FOXP3+ T_regs_ all expressed the inhibitory immune checkpoint VISTA (Figure 1d), with CD45^hi^/CD11b^hi^ cells exhibiting the highest MFI, followed by T_regs_ and lastly by CD45^int^/CD11b^int^ cells (Figure 1e).

### 3.2. TAMs Significantly Infiltrate Medulloblastoma Tumors and Display a Mixed Pro- and Anti-Inflammatory Myeloid-Specific Signature

To further discern immunological markers expressed on TAMs within the tumor microenvironment, we performed CD11b bead enrichment on immune cells isolated from sham control or tumor-bearing mice. Our first aim was to ensure that changes in the myeloid compartment were tumor-specific rather than related to procedural manipulation (*n* = 7 sham, *n* = 24 tumor-bearing mice). We observed global changes between our sham control mice and tumor-bearing mice, with a dramatic increase in the number and percent of CD45^hi^/CD11b^hi^ macrophage-like cells (Q2) (Figure 2a,b). While we also observed a concurrent increase in the number of CD45^int^/CD11b^int^ microglia-like cells, the percentage of CD45^int^/CD11b^int^ relative to the CD45 compartment decreased (Figure 2b). Additionally, we found that, compared to sham controls, CD45^hi^/CD11b^hi^ cells from tumor-bearing mice exhibited upregulated expression of several immune checkpoints, including VISTA, CD80, PD-L1, and CTLA-4 but not Tim-3 (Figure 2c). Among these immune checkpoint molecules, VISTA exhibited the highest level of expression (percent mean ± SEM = 21.32 ± 2.23%), indicating that the expression of this inhibitory immune checkpoint predominates among tumor-associated macrophages (Figure 2c). In contrast, CD45^int^/CD11b^int^ microglia-like cells showed a significant decrease in VISTA expression in tumor-bearing mice compared to sham controls, a phenomenon which has previously been observed in other studies [33]. Beyond immune checkpoints, we also investigated myeloid polarization markers. We found that CD45^hi^/CD11b^hi^ macrophage-like cells exhibited upregulation of MHCII, CD40, CD68, and Arg1, revealing a mixed picture of anti- and pro-inflammatory markers (Figure 2c). To identify whether these markers were unique to infiltrating myeloid cells compared to all other cells in the tumor microenvironment, we compared the CD11b+ cells from the CD11b-positive bead-enriched fraction to CD11b- cells from the CD11b-negative flow-through (*n* = 11). We found that VISTA, CD80, PD-L1, CTLA-4, MHCII, CD40, and CD68 were all enriched in the CD11b+ fraction compared to the CD11b-negative fraction, indicating that this was a unique myeloid signature (Figure 2d). Interestingly, Arg1, CD163, and Tim3 appeared to be enriched on CD11b- cells in a fraction of the samples (Figure 2d).

### 3.3. Myeloid Cells Clustered at the Tumor Border Exhibit Strong VISTA Expression and Ameboid Morphology

To better understand the spatial organization of immune cells in the context of the tumor microenvironment, we performed immunohistochemistry staining for CD11b, VISTA, CD3, and FOXP3 on formalin-fixed paraffin-embedded brains of mCB DNp53 MYC tumor-bearing mice. We observed an abundant accumulation of CD11b+ myeloid cells at the edge of the tumors, many of which co-expressed VISTA (Figure 3a and Appendix A). Furthermore, these myeloid cells exhibited a more ameboid morphology and heightened VISTA expression compared to the ramified cells in the adjacent non-tumor cerebellum (Figure 3a,b). Within the tumor microenvironment, very few CD3+ T-cells were present (Figure 3a). Some of these CD3+ T-cells were also FOXP3+, indicative of T_regs_ (Figure 3a). Intriguingly, the majority of CD3+FOXP3+ T-cells accumulated near perivascular regions rather than in the tumor core or at the tumor edge, appearing trapped (Appendix A). Staining of CD3 and FOXP3 of a naïve wildtype spleen was used as a positive control (Appendix A).

### 3.4. Medulloblastoma Tumor Cells Significantly Inhibit CD4+ T-Cell Proliferation In Vitro

While myeloid cells have been demonstrated to contribute to the immunosuppressive tumor microenvironment, tumor cells have also been shown to promote immune evasion by functionally impairing T-cells [34]. However, this has not been explored in MB. Given that most T-cells within our tumor microenvironment were CD4 T-cells, we co-cultured CD4+ T-cells with mCB DNp53 MYC tumor cells to investigate the direct impact of our tumor cells on T-cell proliferation. We bead-activated naïve splenic CD4+ T-cells with CD3/CD28 beads co-cultured with either EF control medium, tumor cell-conditioned medium, or mCB DNp53 MYC tumor cells (CD4 T-cell/ mCB DNp53 MYC tumor cell ratios: 80:1, 40:1, and 20:1) for five days. We found that co-culturing CD4 T-cells with increasing numbers of mCB DNp53 MYC tumor cells led to a corresponding decrease in T-cell proliferation, with significant inhibition at a T-cell/mCB MYC DNp53 ratio of 20:1, as compared to control medium or tumor-conditioned medium (*n* = 3, *p* = 0.0021) (Figure 4a,b). CD4+ T-cells co-cultured with HEK293T cells at a 20:1 ratio did not show inhibited T-cell proliferation, implying that the observed reduction in proliferation was not a result of cellular crowding or nutrient competition (Figure 4b). To assess whether this observed effect was dependent on VISTA, we added increasing concentrations (0, 0.2, 2, and 20 µg/mL) of VISTA-blocking antibody (clone 13F3) to CD4 T-cells co-cultured with mCB DNp53 MYC tumor cells at a 20:1 ratio (Figure 4c,d). We observed a trend toward increased proliferation with 0.2 µg/mL of 13F3, although this effect was not significant (*p* = 0.4963) (*n* = 6) (Figure 4c,d).

### 3.5. Human and Mouse Medulloblastoma Tumor Cells Express VISTA Binding Partners

To investigate the molecular mechanisms underlying tumor-mediated T-cell inhibition, we examined whether MB tumor cells express either of VISTA’s binding partners implicated in T-cell suppression, VSIG3 and VSIG8 [35,36]. We performed flow cytometry on mCB DNp53 MYC and two of the classic Group 3/4 human MB cell lines, D283-MED and D425-MED. We found that both mouse and human MB cell lines express VSIG3 and VSIG8 in vitro (Figure 5a and Appendix A). This was further confirmed by Western blot analysis (Figure 5b and Appendix A). We also examined VISTA expression and found it to be absent in both human and mouse MB cell lines, indicating that VISTA is predominantly associated with immune cells rather than tumor cells within the context of MB (Figure 5b and Appendix A). Motivated by these findings, we asked whether these binding partners were translationally relevant. Transcriptomic analysis of bulk RNA-seq from human MB utilizing the Caris Life Sciences database revealed a positive correlation between VISTA expression and VSIG8, but no correlation with VSIG3 (Figure 5c). Additionally, we observed a strong correlation between VISTA and two other known binding partners typically expressed on T-cells, PSGL-1 and LRIG1, which we did not have the opportunity to explore further in our mouse model (Figure 5c) [37,38]. Conversely, no correlation was found between VISTA and the housekeeping genes HPRT1 and RPL0 (Appendix A).

### 3.6. Medulloblastoma Exhibits a Unique TAM Phenotype across Human Cohorts Which Is Reflected by the mCB DNp53 MYC Murine Medulloblastoma Model In Vivo

We further queried the Caris transcriptomic database to determine whether markers associated with TAMs in our mouse model could be detected in human MB. We used IBA-1 as a marker for human TAMs. Excitingly, we found that increasing levels of IBA-1 correlated with VISTA, along with other immune checkpoints such as CD80, PD-L1, and CTLA-4 (Figure 6a). Differing from our mouse data, there was a strong correlation between IBA-1 TAMs and Tim-3 (Figure 6a). Additionally, we observed that IBA-1 expression positively correlated with myeloid markers such as HLA-DRA, CD68, and CD40 but only modestly correlated with Arg1 and showed no correlation with the housekeeping gene TBP (Figure 6a and Appendix A). To determine whether these transcriptomic findings correlated at the proteomic level, we utilized Nanostring GeoMx^®^ Digital Spatial Profiling on a human embryonal tumor tissue microarray. We selected 12 intact specimens from different patients, constituting four SHH-activated MBs, five non-WNT/non-SHH MBs, and three MBs of indeterminate subgroups. Immunofluorescent probes stained IBA-1-expressing cells within the tumors. Proteins unique to geometric regions of IBA-1 positivity were isolated using Nanostring GeoMx^®^ rare-cell masking software. Protein expression was quantified to determine differences between the tumors’ IBA-1-positive and -negative areas. IBA-1+ regions were enriched for immune checkpoints such as VISTA, CD80, PD-L1, and CTLA-4, consistent with Caris’ transcriptomic database (Figure 6b). These regions were also enriched for myeloid polarization markers such as HLA-DR, CD68, CD40, and Arg1 (Figure 6b). Distinct from our mouse model but consistent with the human transcriptomic data, Tim-3 was increased across IBA-1+ TAMs in this cohort (Figure 6b).

## 4. Discussion

To the best of our knowledge, this is the first study to thoroughly evaluate the expression of VISTA in the MB tumor microenvironment. We have shown that VISTA is highly expressed across infiltrating immune cells within our syngeneic murine model, including CD45^int^/CD11b^int^ microglia-like TAMs, CD45^hi^/CD11b^hi^ macrophage-like TAMs, and CD4+FoxP3+ Tregs [14]. While all myeloid populations highly express VISTA, we observed decreased expression of VISTA on microglia-like TAMs and an increase in VISTA expression on macrophage-like TAMs, when compared to sham controls, as demonstrated in Figure 2c. This opposing pattern of VISTA regulation has been observed in other CNS pathologies and may indicate that while microglia downregulate VISTA to allow for T-cell infiltration, infiltrating BMDMs can oppose this response [33]. Additionally, microglia may enter an activated macrophage-like state in which they contribute to VISTA-mediated immune suppression along with BMDMs. This aligns with the observed percent decrease in the CD45^int^/CD11b^int^ compartment with a concurrent increase in the CD45^hi^/CD11b^hi^ compartment in tumor-bearing mice as compared to sham control mice, as shown in Figure 2b. Furthermore, we were unable to identify expression of Tmem119 among microglia-like TAMs, aligning with a recent study which used a different mouse MB model, and consistent with the idea that microglia may downregulate Tmem119 and adopt a more macrophage-like phenotype upon entering the tumor microenvironment [32]. High co-expression of VISTA with MHCII was also observed on both myeloid TAM populations, which suggests to us that tumor-infiltrating myeloid cells in MB have heightened antigen presentation capabilities consistent with an activated macrophage phenotype but do so in the context of a novel inhibitory immune checkpoint molecule, VISTA. Importantly, this high co-expression level was not evident in myeloid cells within the bone marrow, underscoring the tumor-specific nature of heightened VISTA and MHCII. These findings are supported by the human mRNA and protein expression profiling where both VISTA and HLA-DR were highly correlated with IBA-1+ infiltrating immune cells, as shown in Figure 6.

Immunohistochemistry on resected murine brain tumors was used to assess spatial localization of cellular populations within the tumors. We found that VISTA-expressing myeloid cells were present throughout murine MB tumors, but the highest VISTA expressors congregated at the tumor border. The presence of these myeloid cells aggregating at the tumor border has been observed in other brain tumor models with low T-cell presence, including MB [32,39]. This spatial distribution along the tumor border has been postulated to form a physical barrier that reduces the activation and infiltration of effector T-cells while promoting a Treg response [40]. In support of this and similar to human MB, across all murine IHC samples, only a small number of CD3+ lymphocytes were detected within the tumor or at the tumor border. Instead, the majority of CD3 staining, though still sparse, was localized to perivascular areas of the tumor. This may indicate that T-cells are trapped in these areas within the tumor, as has been observed in other cold tumors [41]. Several of these CD3+ T-cells stained positive for FOXP3, supporting our flow cytometry data, which detected the limited presence of FOXP3+ T_regs_. These VISTA^hi^-expressing myeloid cells displayed a more ameboid morphology compared to the ramified appearance and weaker VISTA staining of myeloid cells in the adjacent non-tumor cerebellum and remote cortical regions, as demonstrated in Figure 3b. This supports the idea that these high-expressing VISTA cells may be the opposing force responsible for preventing immune infiltration into the tumor. Unfortunately, we were unable to confirm these findings in our human samples. The use of minimally invasive neurosurgical techniques has improved neurocognitive outcomes by carefully removing tumors from critical brain regions, but this impedes evaluation of the peritumoral border in most diagnostic samples [42]. Nevertheless, these novel observations in our mouse model support our earlier findings that human MB exhibits poor T-cell infiltration and help explain its poor response to T-cell-targeting immune checkpoint blockade [11]. Taken together, we believe these findings support the notion that T-cell infiltration and antigen response mediated by CD45^int^/CD11b^int^ microglia-like cells are impeded by the presence of CD45^hi^/CD11b^hi^ macrophage-like TAMs through a VISTA-dependent mechanism in our mouse model. These findings hold translational implications, as they highlight the importance of targeting myeloid cells within the tumor microenvironment of MB and further uncovering the role VISTA may play in these cells by promoting T-cell suppression.

Negative effects on T-cells were also reflected by the remarkable ability of mCB DNp53 MYC tumor cells to override the exogenous solid activation signals provided by CD3/CD28 combined with IL-2 in vitro. These experiments demonstrated for the first time that MB tumor cells can function as direct CD4 T-cell suppressors. This suppression of T-cell proliferation was observed exclusively in the presence of tumor cells rather than soluble factors from conditioned media. This implies a direct interaction between tumor cells and CD4+ T-cells operating independently of cytokine-mediated pathways. It also underscores the likelihood that MB employs multiple mechanisms of immune suppression. Therefore, even if immune modulatory therapies allow T-cells to penetrate the tumor, there will likely be additional hurdles to the persistence of T-cells in this microenvironment due to the significant capacity of tumor cells to hinder T-cell proliferation. These findings have profound implications for therapies such as CAR T-cells, which need to be able to proliferate and expand upon entry into the tumor microenvironment. They also have implications for immune checkpoint therapies, which may not be able to overcome the strong immunosuppressive signals delivered by tumor cells. This was demonstrated in vitro by the inability of VISTA-blocking antibodies to significantly rescue T-cells from tumor-mediated suppression of proliferation. Therefore, while VISTA may be an important mediator of immune suppression in MB, combination therapies will likely be needed to fully overcome the powerful immunosuppressive effects mediated by this tumor.

Identifying the specific mechanism of VISTA-mediated T-cell inhibition in MB was beyond the scope of this study. However, we identified multiple VISTA binding partners in human and murine cell lines as well as in our human cohorts where VISTA signaling likely occurs. Both described tumor-expressed binding partners, VSIG3 and VSIG8, were expressed in vitro. Transcriptomic analysis of human MB tumors revealed that VISTA expression positively correlated with VSIG8 but not VSIG3, suggesting that VSIG8 may be a more translationally relevant target. Moreover, the low expression of VISTA on our tumor cells indicated that VISTA expression is predominantly confined to immune cells within this tumor. This finding aligns with previous studies showing that VISTA is typically exclusive to either tumor cells or immune cells within the tumor microenvironment [43]. Multiple binding partners on tumor, myeloid, and T-cells have been described for VISTA in various contexts. Two of these, PSGL-1 and LRIG1, also correlated with VISTA expression in human transcriptomic data, as shown in Figure 5. It will be important to understand which specific binding partner(s) are most relevant in MB for more effective targeting of the VISTA signaling axis. Identifying and investigating the VSIG family in MB is a novel aspect of this work and deserves further consideration in future studies. Whether these binding partners actively engage VISTA in this tumor microenvironment remains unclear.

We recognize several limitations of the current study. Given the paucity of infiltrating immune cells in MB, one major constraint was the limited number of cells isolated from tumors. Despite efforts to increase immune cell recovery through bead enrichment, only a few tens of thousands of total immune cells could be isolated from a single tumor. This challenge extended to the Caris human sequencing database due to the large abundance of tumor cells and the low number of infiltrating immune cells, exacerbated by a small cohort (*n* = 52). Additionally, distinguishing microglia from bone marrow-derived macrophages within our flow cytometry data proved challenging, as both upregulate CD45 and CD11b. Conventional markers for microglia, such as Tmem119, were absent in our model. The mixed activating and inhibitory immune signature exhibited by TAMs in both human and mouse MB underscored the difficulty of delineating these states in microglia and macrophages. Despite the overwhelming similarities we observed between our human cohorts and our mouse model, there were some notable differences between mice and humans, including Arg1, CD163, and Tim-3. These markers are not usually associated with tumor cells and were only upregulated in a fraction of the samples. Whether this represents tumor heterogeneity within our mouse model or is an artifact of the bead isolation procedure remains unclear, and further studies focusing on these markers are needed. Together, these issues highlight the limitations of using surface markers in the setting of tumors. Thus, single-cell analysis may offer a more sensitive approach to the immune phenotyping of these complex and dynamic immune cells. Of note, our flow cytometry gating of myeloid subpopulations excluded CD11c cells—a widely established pan-dendritic marker. Given that CD11c is a marker that was upregulated in our human IBA-1+ TAMs as well, it may be that CD11c+ TAMs exist in MB tumors and have a function that has yet to be investigated [39].

## 5. Conclusions

Overcoming the immunosuppressive tumor microenvironment of MB remains a significant hurdle for immunotherapies to achieve successful clinical outcomes. Thus far, no meaningful clinical improvement has been seen in clinical trials evaluating any immune-mediated strategy. Our study unveils a multifaceted landscape of immune-evasion mechanisms that attempt to explain the scarcity of lymphocytes within the tumor microenvironment. We show an overwhelming presence of myeloid cells expressing the negative checkpoint regulator VISTA. Our emphasis on the disparate regulation of VISTA across myeloid subpopulations indicates that microglia-like and macrophage-like cells may play opposing roles in the tumor microenvironment and serves as a compelling rationale to further explore their exact roles. Moreover, the physical barrier of VISTA-positive myeloid cells at the tumor periphery suggests a pivotal myeloid–tumor interplay, while the inhibition of T-cell proliferation by tumor cells underscores a T-cell–tumor inhibitory effect driving immune suppression. Furthermore, identifying VISTA binding partners on tumor cells carries significant translational implications and supports the idea of targeting the VISTA–VSIG axis to enhance anti-tumor responses. We advocate the adoption of combination immunotherapy that includes targeting the VISTA signaling axis as a robust strategy to counteract the myriad of mechanisms contributing to immune suppression in the hope of advancing treatment for patients with MB.

## Figures and Tables

**Figure 1 cancers-16-02629-f001:**
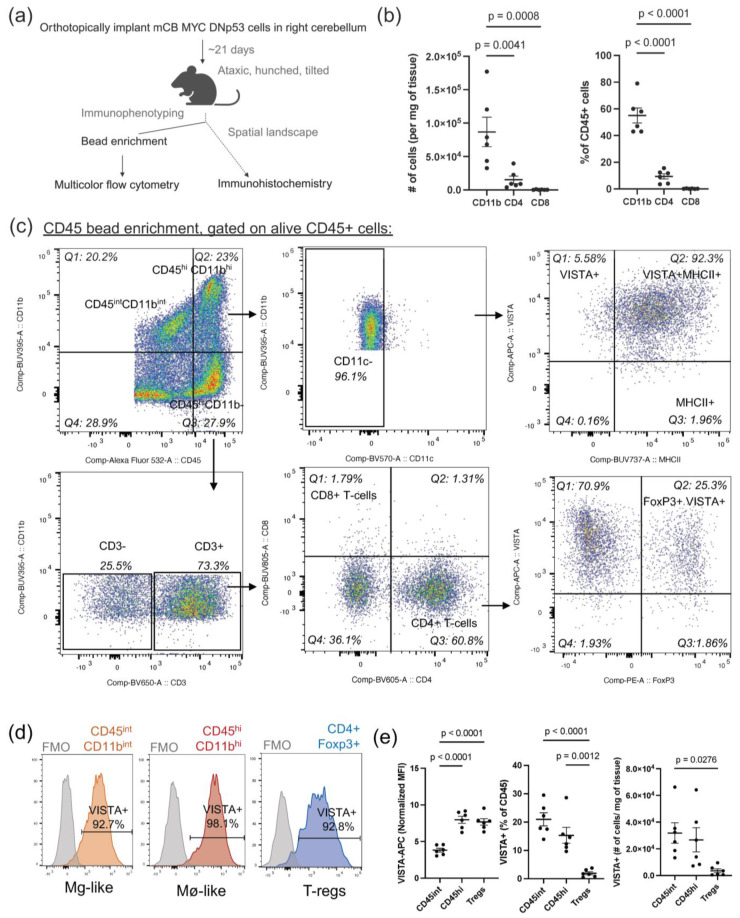
Syngeneic mouse model of MB exhibits expression of VISTA on tumor-infiltrating myeloid cells and T_regs_. (**a**) Experimental scheme of C57BL/6J mouse orthotopically implanted with mCB DNp53 MYC tumor cells and the downstream workflow to immune-phenotype the tumor microenvironment. (**b**) Following CD45 bead enrichment of dissociated tumors, populations of myeloid and lymphoid cells were analyzed across 6 tumor-bearing mice using flow cytometry. Percent of CD45 and number of cells per mg of tumor shown for CD11b myeloid cells, CD4 T-cells, and CD8 T-cells (**c**) Representative flow cytometric analysis of tumor-infiltrating immune cells following CD45 bead enrichment. Gating performed on live CD45+ singlets. Myeloid and lymphoid subpopulations and their corresponding VISTA expressions. (**d**) Histograms displaying percent of VISTA-APC expression within the CD45^int^/CD11b^int^/CD11c-, CD45^hi^/CD11b^hi^/CD11c-, and CD4+FOXP3+ compartments compared to fluorescence minus one (FMO) controls. (**e**) Normalized MFI ((MFI-control MFI)/control MFI), percent of CD45, and number of cells per milligram of tumor of VISTA-positive cells across CD45^int^/CD11b^int^/CD11c-, CD45^hi^/CD11b^hi^/CD11c- cells, and CD4+FOXP3+ T_regs_ for all replicates. For (**b**–**e**), *n* = 6 tumor-bearing female mice. Error bars represent standard error of the mean (SEM), and *p*-values were calculated by ordinary one-way ANOVA. Only significant values shown.

**Figure 2 cancers-16-02629-f002:**
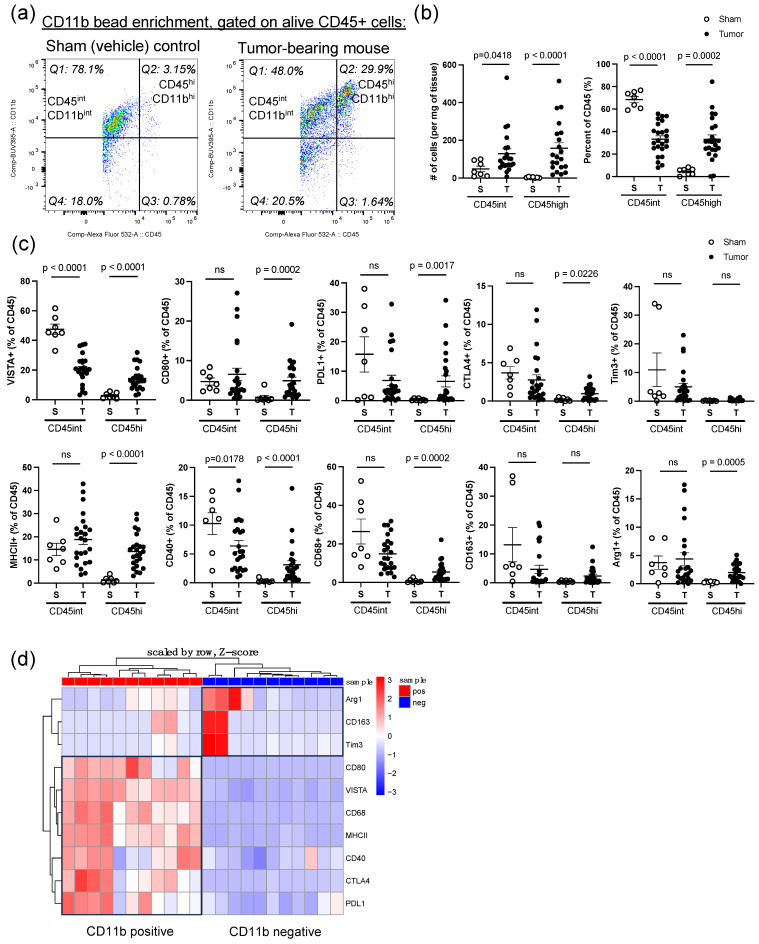
CD45^hi^/CD11b^hi^ macrophage-like cells significantly infiltrate tumors and display a mixed pro- and anti-inflammatory signature that is unique to myeloid cells in the tumor microenvironment. (**a**) Representative flow cytometric analysis of CD11b bead-enriched immune cells in the right cerebellum of sham-injected mice (*n* = 7; 7 females) vs. tumors of tumor-bearing mice (*n* = 24; 19 females, 5 males), both sacrificed 21 days post-implantation. Gating performed on live CD45+ singlets. (**b**) Comparison of the percent of total CD45 and number of cells per milligram (mg) of tissue for CD45^int^/CD11b^int^ microglia-like cells and CD45^hi^/CD11b^hi^ macrophage-like cells between sham control and tumor-bearing mice. (**c**) Comparison of different immune checkpoints (VISTA, CD80, PDL1, CTLA4, and Tim-3) and myeloid polarization markers (MHCII, CD40, CD68, Arg1, and CD163) between sham control (open dot) and tumor-bearing mice (closed dot) for CD45^int^/CD11b^int^ microglia-like cells and CD45^hi^/CD11b^hi^ macrophage-like cells, displayed as percent of total CD45 cells. (**d**) Heatmap comparing percent of immune markers on CD11b-positive cells from the CD11b bead-enriched fraction (red) to CD11b-negative cells from the CD11b-negative flow-through (blue) (*n* = 11; 10 females, 1 male). Markers were considered differentially expressed between compared groups with a log2 fold change less than −0.5 or greater than 0.5 and a *p*-value less than 0.05. For (**b**,**c**), error bars represent standard error of the mean (SEM), and *p*-values were calculated using a Mann–Whitney test. ns = not significant.

**Figure 3 cancers-16-02629-f003:**
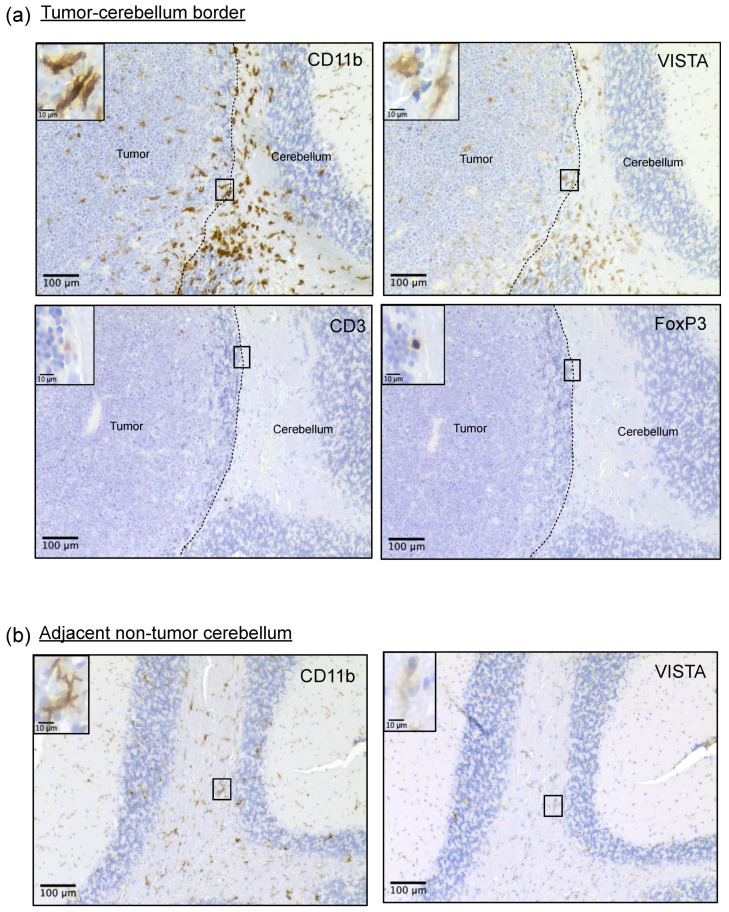
Myeloid cells clustered at the tumor edge exhibit strong VISTA expression and ameboid morphology. (**a**) Whole brains from tumor-bearing mice were grossly dissected, paraffin-embedded, sectioned, and analyzed by immunohistochemistry. Consecutive tissue sections were stained for CD11b, VISTA, CD3, and FOXP3. Images represent 200× magnifications and highlight tumor borders, with the tumor on the left and the adjacent non-tumor cerebellum on the right; scale bar represents 100 μm. Inset highlights specific cells, scale bar represents 10 μm. (**b**) CD11b and VISTA expression in adjacent non-tumor cerebellum. Image represents 200×; scale bar represents 100 µm. Inset highlights specific cells; scale bar represents 10 µm.

**Figure 4 cancers-16-02629-f004:**
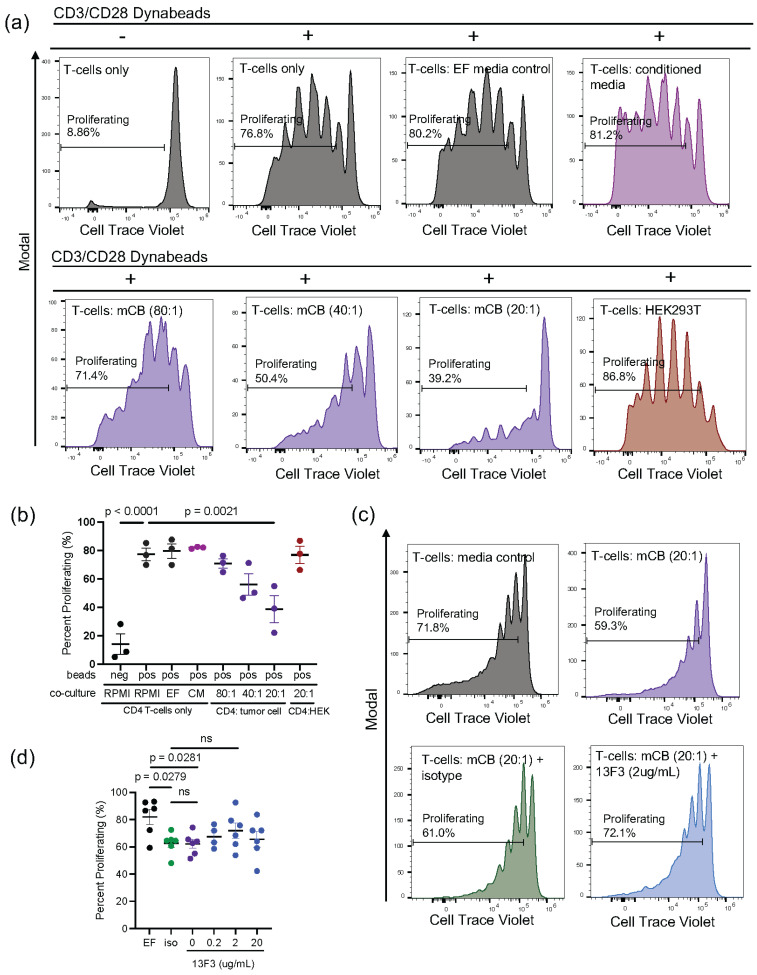
mCB DNp53 MYC tumor cells significantly inhibit CD4+ T-cell proliferation in vitro. (**a**) Naïve splenic CD4 T-cells activated with CD3/CD28 beads and recombinant IL-2 stained with CellTrace-Violet and co-cultured for 5 days either alone, with EF control medium, conditioned medium, mCB DNp53 MYC tumor cells at increasing CD4/mCB DNp53 MYC ratios of 80:1, 40:1, and 20:1, or HEK293T cells (CD4/HEK283T ratio 20:1). CellTrace-Violet dilution is measured as a function of proliferation; representative histograms are shown. Cells gated on live non-GFP CD4+ T-cells. Negative control was T-cells co-cultured without CD3/CD28 beads. (**b**) Percent of CD4+ T-cell proliferation analyzed across 3 independent experiments using an ordinary one-way ANOVA. (**c**) CD4 T-cells and mCB DNp53 cells co-cultured at a ratio of 20:1 treated with 2 μg/mL of anti-VISTA blocking antibody (clone 13F3) or isotype control (Armenian Hamster IgG). (**d**) Percent of CD4+ T-cell proliferation was analyzed across 6 independent experiments using an ordinary one-way ANOVA. Drug concentrations included 0 μg/mL, 0.2 μg/mL, 2 μg/mL, and 20 μg/mL of clone 13F3 or 20 μg/mL of Armenian Hamster IgG isotype control. For (**d**), each independent experiment was performed in technical triplicates and values were averaged. Error bars represent standard error of the mean (SEM), and *p*-values were calculated using a one-way ANOVA. ns = not significant.

**Figure 5 cancers-16-02629-f005:**
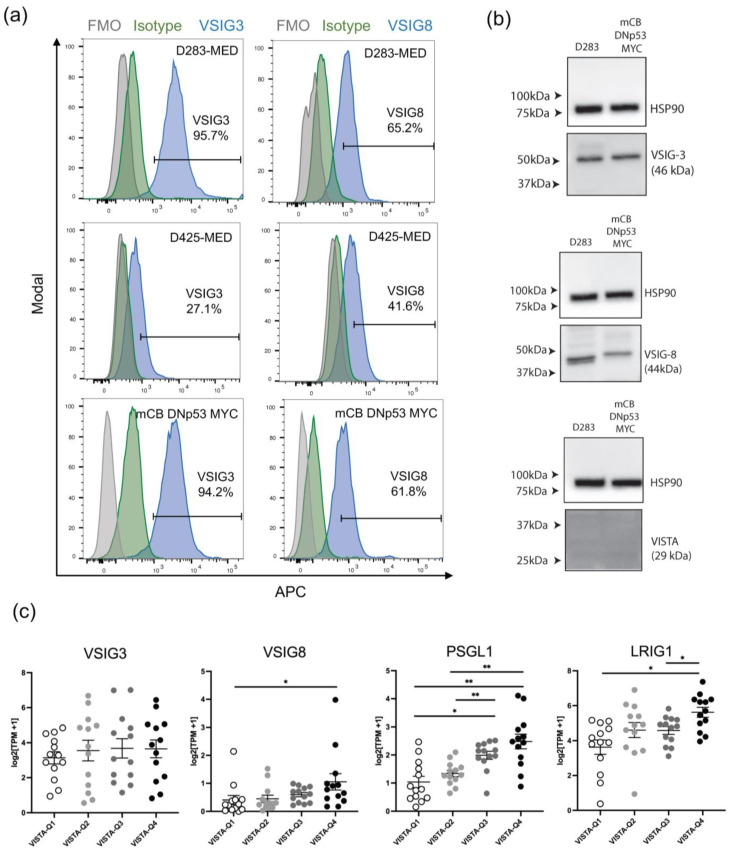
Human and mouse MB tumor cells exhibit in vitro expression of VISTA binding partners and are translationally relevant. (**a**) Cells gated on live singlets, then further gated based on GFP expression. Histograms (blue) representing in vitro expression of VSIG8 and VSIG3 in human MB cell lines (D283-MED and D425-MED) and our GFP+ mouse medulloblastoma cell line (mCB DNp53 MYC). Fluorescence minus one control represented in gray, and isotype control represented in green. (**b**) Western blot analysis confirming expression of VSIG3 and VSIG8 and relative absence of VISTA from human and mouse medulloblastoma tumor cell lines (*n* = 3). Uncropped blots are shown in Appendix A. (**c**) Transcriptome analysis of 52 human MB patient samples. *X*-axis represents quartile distribution of VISTA expression levels: Q1 (lowest 25%), Q2 (next 25%), Q3 (third 25%), and Q4 (highest 25%). *Y*-axis represents log2 [TPM+1]. Bar graphs representing VISTA transcript (C10orf54) expression correlation with VSIG3, VSIG8, and PSGL1 and LRIG1. Error bars represent standard error of the mean (SEM), and *p*-values were calculated using a Mann–Whitney test and corrected for multiple comparisons. The asterisks indicate adjusted *p*-values as follows: * padj < 0.05, ** padj < 0.01.

**Figure 6 cancers-16-02629-f006:**
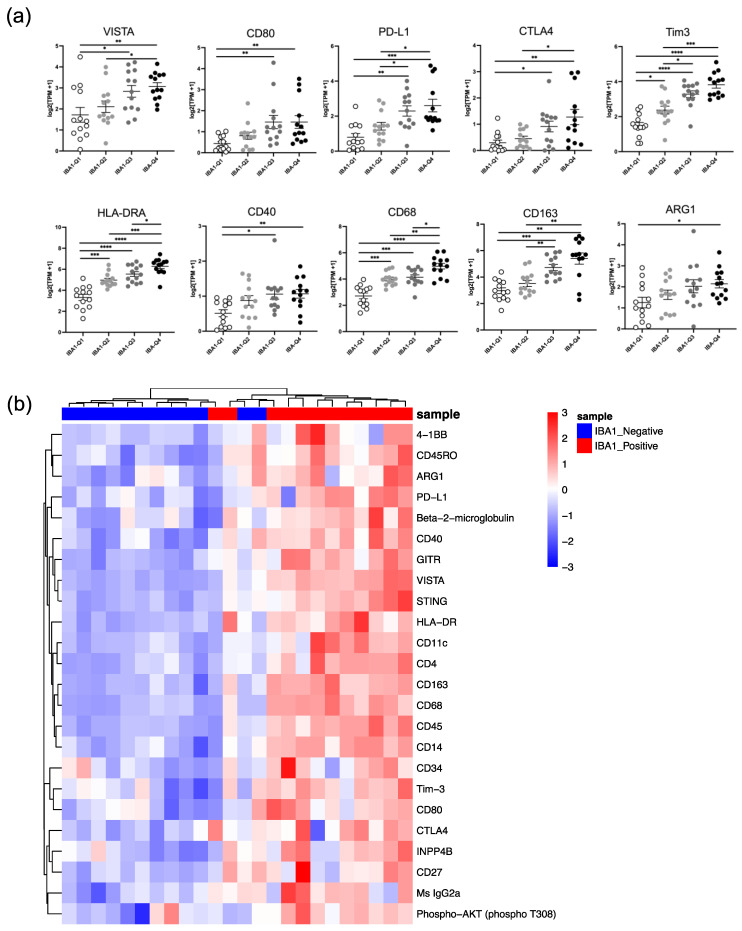
The mCB MYC DNp53 mouse medulloblastoma model faithfully recapitulates human disease. (**a**) Transcriptome analysis of 52 human MB patient samples using the Caris Life Sciences database. Quartile expression of IBA-1 transcript (Aif1) was correlated to immune checkpoint and myeloid polarization markers found in our mouse MB model. *X*-axis represents quartile distribution of IBA-1 transcript expression levels: Q1 (lowest 25%), Q2 (next 25%), Q3 (third 25%), and Q4 (highest 25%). *Y*-axis represents log2 [TPM+1]. (**b**) Nanostring GeoMx^®^ Digital spatial profiling performed on a human tissue microarray of MB differential protein expression of immune markers in IBA-positive cells was compared to all other IBA-1-negative cells. Proteins were considered differentially expressed between compared groups with a log2 fold change less than −0.5 or greater than 0.5 and a *p*-value less than 0.05. In (**a**), error bars represent standard error of the mean (SEM), and *p*-values were calculated using a Mann–Whitney test and corrected for multiple comparisons. The asterisks indicate adjusted *p*-values as follows: * padj < 0.05, ** padj < 0.01, *** padj < 0.001, **** padj < 0.0001.

## Data Availability

The datasets generated and/or analyzed during the current study are available from the corresponding author on reasonable request. The deidentified sequencing data cannot be publicly shared due to the data usage agreement between the facilities of the study team. Data will be made available upon reasonable request with the permission of Caris Life Sciences.

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
