# Peer review of "VISTA Emerges as a Promising Target against Immune Evasion Mechanisms in Medulloblastoma"

_cancers, 2024, doi:10.3390/cancers16152629_

Round 1

Reviewer 1 Report

Comments and Suggestions for Authors

The authors carried out a molecular study to identify a biotarget in medulloblastoma, a very aggressive malignant tumour that is common in children. They explore an area involving the immune system and identify the negative checkpoint regulator VISTA, which is involved in the phenomenon of immunosuppression.  The manuscript is comprehensive in all areas and the results were obtained using appropriate experimental methods. Through this study, the authors provide useful results in the search for suitable drugs for medulloblastoma. All figures, especially the lettering, are blurred and should be improved.

Author Response

Comments1: 

The authors carried out a molecular study to identify a biotarget in medulloblastoma, a very aggressive malignant tumour that is common in children. They explore an area involving the immune system and identify the negative checkpoint regulator VISTA, which is involved in the phenomenon of immunosuppression.  The manuscript is comprehensive in all areas and the results were obtained using appropriate experimental methods. Through this study, the authors provide useful results in the search for suitable drugs for medulloblastoma. All figures, especially the lettering, are blurred and should be improved.

Response 1: 

We thank the reviewer for reviewing our manuscript and for the positive comments on our work.  We regret that the figures were blurry when interfaced with the word document.  We have carefully reintegrated the original figures with the text and believe that this problem is now fixed.  Thank you for bringing this to our attention.

Reviewer 2 Report

Comments and Suggestions for Authors

I read and reviewed the paper titled "VISTA emerges as a promising target against immune evasion mechanisms in medulloblastoma", carefully. Authors studied the mechanisms behind the low T-cell infiltration in relapsed medulloblastoma (MB) and identify potential therapeutic targets to improve the efficacy of immunotherapies. The study uses a syngeneic mouse model to examine the tumor immune microenvironment and compares the findings to human MB data, highlighting the role of tumor-associated macrophages (TAMs), regulatory T cells (Tregs), and the inhibitory checkpoint molecule VISTA. The research aims to uncover the interactions between VISTA and its binding partner VSIG8, proposing that targeting this axis could enhance anti-tumor immune responses in MB. My comments are as follows:

- VISTA should be spelled in the title. No abbreviations in the title please.

- Minor typos must be fixed up. (i.e., CO2 instead of CO2).

- Introductory, methodology and the results sections are flawless. Adequate background data and clear objectives were provided in the introduction. Methods and statistical analyses were comprehensively expressed in the methodology. Presentation of the results was very well with adequate figures. However, discussion could be improved. The first paragraph of the discussion must summarize the main outcomes of the work. Moreover, clinical use of the findings in daily practice should be commented on in the discussion.

Comments on the Quality of English Language

Dear Editor

I read and reviewed the paper titled "VISTA emerges as a promising target against immune evasion mechanisms in medulloblastoma", carefully. Authors studied the mechanisms behind the low T-cell infiltration in relapsed medulloblastoma (MB) and identify potential therapeutic targets to improve the efficacy of immunotherapies. The study uses a syngeneic mouse model to examine the tumor immune microenvironment and compares the findings to human MB data, highlighting the role of tumor-associated macrophages (TAMs), regulatory T cells (Tregs), and the inhibitory checkpoint molecule VISTA. The research aims to uncover the interactions between VISTA and its binding partner VSIG8, proposing that targeting this axis could enhance anti-tumor immune responses in MB. My comments are as follows:

- VISTA should be spelled in the title. No abbreviations in the title please.

- Minor typos must be fixed up. (i.e., CO2 instead of CO2).

- Introductory, methodology and the results sections are flawless. Adequate background data and clear objectives were provided in the introduction. Methods and statistical analyses were comprehensively expressed in the methodology. Presentation of the results was very well with adequate figures. However, discussion could be improved. The first paragraph of the discussion must summarize the main outcomes of the work. Moreover, clinical use of the findings in daily practice should be commented on in the discussion.

Author Response

Comments 1: I read and reviewed the paper titled "VISTA emerges as a promising target against immune evasion mechanisms in medulloblastoma", carefully. Authors studied the mechanisms behind the low T-cell infiltration in relapsed medulloblastoma (MB) and identify potential therapeutic targets to improve the efficacy of immunotherapies. The study uses a syngeneic mouse model to examine the tumor immune microenvironment and compares the findings to human MB data, highlighting the role of tumor-associated macrophages (TAMs), regulatory T cells (Tregs), and the inhibitory checkpoint molecule VISTA. The research aims to uncover the interactions between VISTA and its binding partner VSIG8, proposing that targeting this axis could enhance anti-tumor immune responses in MB. My comments are as follows:

Response 1 : We thank the reviewer for such a thorough review of our manuscript.

Comments 2 : - VISTA should be spelled in the title. No abbreviations in the title please.

Response 2 : Thank you, this has been added to the title as V-domain Ig Suppressor of T-cell Activation (VISTA).

Comments 3 : - Minor typos must be fixed up. (i.e., CO2 instead of CO2).

Response 3 :  Thank you, this has been corrected.  We have also carefully profread the manuscript.

Comments 4 : - Introductory, methodology and the results sections are flawless. Adequate background data and clear objectives were provided in the introduction. Methods and statistical analyses were comprehensively expressed in the methodology. Presentation of the results was very well with adequate figures. However, discussion could be improved. The first paragraph of the discussion must summarize the main outcomes of the work. Moreover, clinical use of the findings in daily practice should be commented on in the discussion.

Response 4 : We thank the reviewer for the very positive comments on introduction, methods, results and figure presentation in our manuscript.  We have revised the discussion to begin with a synopsis of findings.  We have added more text to address translational relevance.